# Proposal for Flood Risk Mitigation in the Upper Tanaro Valley (Western Alps—North-Western Italy)

**Battista Taboni ***, **Michele Licata** , **Victor Buleo Tebar, Mauro Bonasera** and **Gessica Umili**

Department of Earth Sciences, University of Turin, Via Valperga Caluso, 35, 10125 Turin, Italy;
michele.licata@unito.it (M.L.); victor.buleotebar@unito.it (V.B.T.); mauro.bonasera@unito.it (M.B.);
gessica.umili@unito.it (G.U.)
* Correspondence: battista.taboni@unito.it

**Abstract:** Flood risk in Italy is a key aspect for the administrative authorities, from the national to the local level. This is especially true in Northern Italy, where the Po River, the most important river of the peninsula, and its river basin are located. In North-Western Italy, the Po Basin is described by numerous sub-basins, among which is the Tanaro River basin: here, in the last decades, floods have produced significant damage, causing an increased concern to local and regional administrations. The main goal of this study was to identify suitable sites for the construction of dams, having the function of retention basins, aiming to mitigate the flood risk in the Upper Tanaro Valley. First, using a qualitative approach, suitable sites were identified using available public data provided by regional administrations and field data obtained from geomorphological surveys, later elaborated in a Geographic Information System (GIS) environment. Several models were then produced using conventional methods to evaluate the hydrological characteristics of the study area and to assess the efficiency of each site in terms of flood water flow rate reduction: the performance was evaluated at control sections chosen in urban areas along the Upper Tanaro Valley. The results show that it is possible to find suitable locations for risk mitigation structures. These models also allowed for a rapid cost-effectiveness evaluation, which led to the definition of the best-performing site. The Upper Tanaro Valley case study here analyzed contributes to proving the importance of an integrated approach based on geomorphological, geo-hydrological, and hydraulic evaluations when dealing with the choice of a flood risk mitigation strategy.

**Keywords:** geo-hydrological risk; floods; risk mitigation; geomorphology; Western Alps



## 1. Introduction

In mountainous areas, the fact that population density is greater along rivers running in the valley bottom is a serious concern worldwide, since the presence of a river is always associated with floods and other extreme events. This problem is enhanced by the morphology of the valley, especially in the case of steep slopes and deeply incised valleys, and by brief and intense rainfall. In Italy this is a key aspect for the administrative authorities, from the national to the local level, as shown by [1]. This is especially true in the northern part of Italy, where the Po River, the most important river of the peninsula, and its basin are located: due to the great length of the Po River, an estimated 4 million people live within the area where the floods are likely to occur [1].

In the north-western portion of northern Italy, the Po Basin is described by numerous sub-basins, among which is the Tanaro River basin. According to [1], the flood activity of the Tanaro River seems to be limited or very limited, but the main residential and productive areas of the Upper Tanaro Valley (UTV) have been subjected, in the period between 1990 and 2021, to a number of floods that have caused serious damage [2], the most recent of which occurred in late October 2020. The situation of the area surrounding the town of Ceva (North-Western Italy, Province of Cuneo) was analyzed by the authors

of [3]: the town itself is associated with a high hazard level related to floods, due to the position of the structures and edifices which are located near or very near the Tanaro River. Another significant problem, pointed out in the cited study, is that the section of the stream, measured in key points along the river, lacks the capacity to guarantee that the water flowing through it is completely contained, even for events with a small recurrence time (i.e., 20 years). This context is well aligned with European data, which identify floods as the most common natural disaster and the one causing the largest number of casualties and economic damage [4]. For this reason, the European Commission and the European Union Council put forward Directive 60/2007/EC [5], aiming to manage and reduce the risk arising from floods and other hazardous processes associated with rivers and water bodies, such as debris flows. If we consider the enhancing effect related to climate change, such as more intense and frequent rainfall events, and the effects of uncontrolled land use [6,7], easily seen in the widespread distribution of buildings in the areas surrounding the Tanaro River, then it becomes clear that there is a need for tools that may allow local and regional administrations to control these floods and prevent major damage, as shown in previous instances of the problem, mitigating the hydrological and geo-hydrological risk of the UTV. These operations are also called Disaster Risk Reduction (DRR) [8]. It should also be taken into account that the Tanaro River is the second longest river of the Piemonte Region (Northwestern Italy), second only to the Po River, of which it is a tributary. Therefore, controlling its flow of water in dangerous situations along its upper portion means easing the consequential effects in the downstream portion, where there is a confluence of the Tanaro River and the Po River.

From this perspective, the need for a simple yet effective method to identify suitable sites to accommodate DRR structures is of great importance to both local and regional administrations. These structures should combine the minimum environmental and aesthetic impact, with the highest risk reduction factor during floods. The opportunity to discriminate between potentially suitable sites in the initial feasibility study, based on their efficiency, can substantially speed up the design phase.

In this study, we propose a preliminary approach for the identification of suitable sites for the construction of DRR structures using a multidisciplinary method. We have applied it to the tributaries in the uppermost portion of the UTV, with the prospect of accommodating the construction of dams having the function of retention basins: by controlling floods in this portion of the analyzed basin it is possible to significantly reduce the risk to which most of the residential and productive areas of the valley are exposed. The area of the uppermost portion of the UTV has been chosen due to different factors: the limited space available for possible DRR implementation, and the need for protecting all the main urbanized and industrial areas of the valley. A key aspect was the fact that the major hydrological sub-basins within the Tanaro basin, called Tanarello Valley and Negrone Valley (following the name of the streams that flow along them), are located right at the uppermost limit of the basin itself. By mitigating the effects of the water flow originating in these sub-basins, significant mitigation of the floods of the Tanaro River can be achieved.

The specific sites have been chosen with regard to a set of qualitative and quantitative parameters, which maximize the beneficial effect of the proposed flood management structures while also minimizing their geomorphological, environmental and visual impact. It must be noted how, in recent years, different and more environmentally sustainable solutions have been proposed and used (usually called Nature-based Solutions, or NBSs) [8,9]. Still, in the context of the valley considered in this study, they were deemed insufficient, especially keeping in mind that these solutions are generally efficient only for small or moderate floods [10]. It should be taken into account that the integration (also called hybrid NBSs [9]) of these new, more sustainable methods with old traditional flood management structures is always a good practice, capable of enhancing their global DRR effect [8].

Once the sites were identified, the mitigation effect of the proposed structures was quantified by evaluating the maximum discharge produced by the hydrological basins. This parameter was assessed in five sections along the Tanaro River, between the towns of

Ormea and Ceva, with and without the DRR structures: several different possible scenarios have been produced and evaluated. Lastly, we have analyzed the results and proposed the most effective solution, always keeping in mind the obvious need for cost-efficiency that both this kind of structure and the administrations managing the territory require.

The method is summarized in the work flow diagram of Figure 1, while in Figure 2 the general geographic setting of the study area is shown.

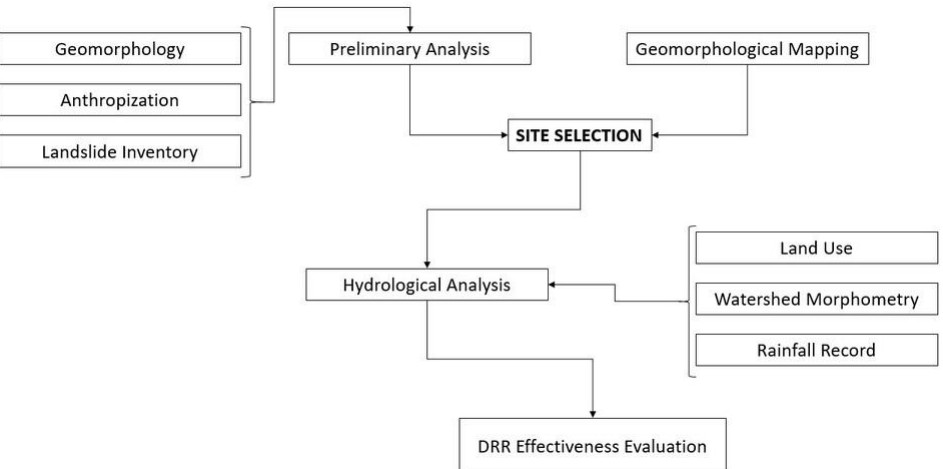

**Figure 1.** Simplified work flow diagram of the method proposed in this study.

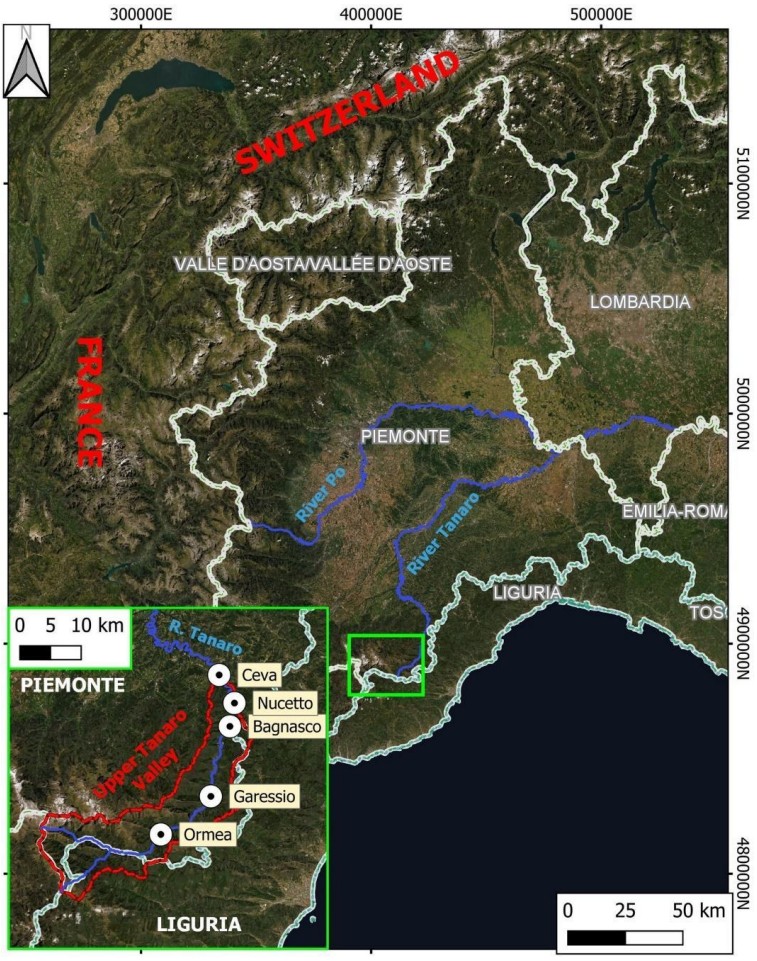

**Figure 2.** Geographical setting of the study area (green rectangle), located in NW Italy.

## 2. Study Area

The area of interest for this study is positioned along the southern termination of the western Alps, partly in the Piemonte Region to the north, where most of the study area is located, and in the Liguria Region to the south (Figure 2).

The southern portion of the Piemonte Region corresponds to the "southern termination of the Western Alps" [11], a region well studied from the geological point of view in recent decades [12]. The structural setting is characterized by a double vergent structure at regional scale with the Briançonnais Domain in the internal sector and the Dauphinois-Provençal Domain in the external one [13]. Several tectonic units are involved in NW-SE strike deformation zones in this area [11], but there are only a few of them in the study area: for a detailed lithostratigraphic description see [12–15].

The study area lies within the upper portion of the Tanaro River drainage basin, with its highest point at Mt. Marguareis (2651 m a.s.l.) and the lowest within the town of Ceva (386 m a.s.l.), covering an area of approximately 400 km$^2$. The uppermost portion of the UTV is divided in two sub-basins, belonging respectively to the Negrone River, which corresponds to the westernmost part of the Tanaro River, and the Tanarello River: from their confluence, the Tanaro River originates. The fluvial dynamic and the intense erosion have produced a rather homogeneous morphology characterized by topography of steep terrain with a deep V-shaped valley, especially in the upper Tanaro River catchments, which this study is considering. Steepness should be deemed as a crucial parameter for any intervention, as well as other topographic criteria such as curvature or landslide distribution: for the Tanarello Basin, 46.8% of the area corresponds to slope values between 20° and 30°, while the Negrone Basin has 67.1% of its surface area between 20° and 40°, being the catchment with the most significant fraction of area with slope angle higher than 50°. There is a substantial relationship between altitude and steepness, especially regarding the right side of the valley, characterized by the presence of large vertical walls. The left side of the valley has a similar morphology but it is reduced and concentrated on higher elevation ranges.

The hydrographic network is generally dominated by SW-NE-directed streams, except for those of a minor order, most of which are direct tributaries of the Tanaro River. The Tanarello River is distinguished by a deep gorge carved into the bedrock in its final stretch before flowing into the Negrone River. Along the valley where the Negrone River is located, some rock ledges appear, spreading from the mid-elevation ranges towards the stream, narrowing down the valley floor and constraining the course of the river.

Land cover is dominated by broad-leaved and mixed forest, primarily located in areas below 1500 m a.s.l. Tanarello and Negrone catchments are characterized by large areas of broad-leaved, mixed and coniferous forests, and to a lesser extent, sparsely vegetated areas as the surface ascends to the highest altitude ranges. Moors, heathlands, natural grasslands, and transitional woodland-shrub areas are also present but cover small areas. Lands principally occupied by agriculture and cultivation areas are also affected by rotational and translational landslides, especially at the end of the valley, close to the valley bottom and, therefore, Tanaro River. Lastly, only a minimal fraction of the study area is occupied directly by rock outcrops.

From a geomorphological point of view, different landslide types are present and therefore controlled by different combinations of environmental and topographic parameters. Most of them belong to rotational and translational landslides, and they are concentrated, in conjunction with shallow phenomena, where lithologies such as marls, sandstones, and conglomerates emerge. Tanarello and Negrone catchments are typically affected by rockfalls and debris flows due to steep slopes and lithologies, such as highly fractured sandstones and limestones. In general, the landslides recorded on this portion of the Tanaro Basin are associated with the 2016 and 2020 floods [2], characterized by brief but extremely intense precipitation [2,3]. In Tanarello and Negrone valleys karst phenomena are relatively common: a wide and complex network of caves and tunnels is known, especially along the north-western portion of the area near the sector of Mt. Marguereis. Data regarding

karst phenomena are incomplete at best, and therefore their consequences on the water circulation in the study area have been mostly ignored.

Figure 3 summarizes the information discussed in this section.

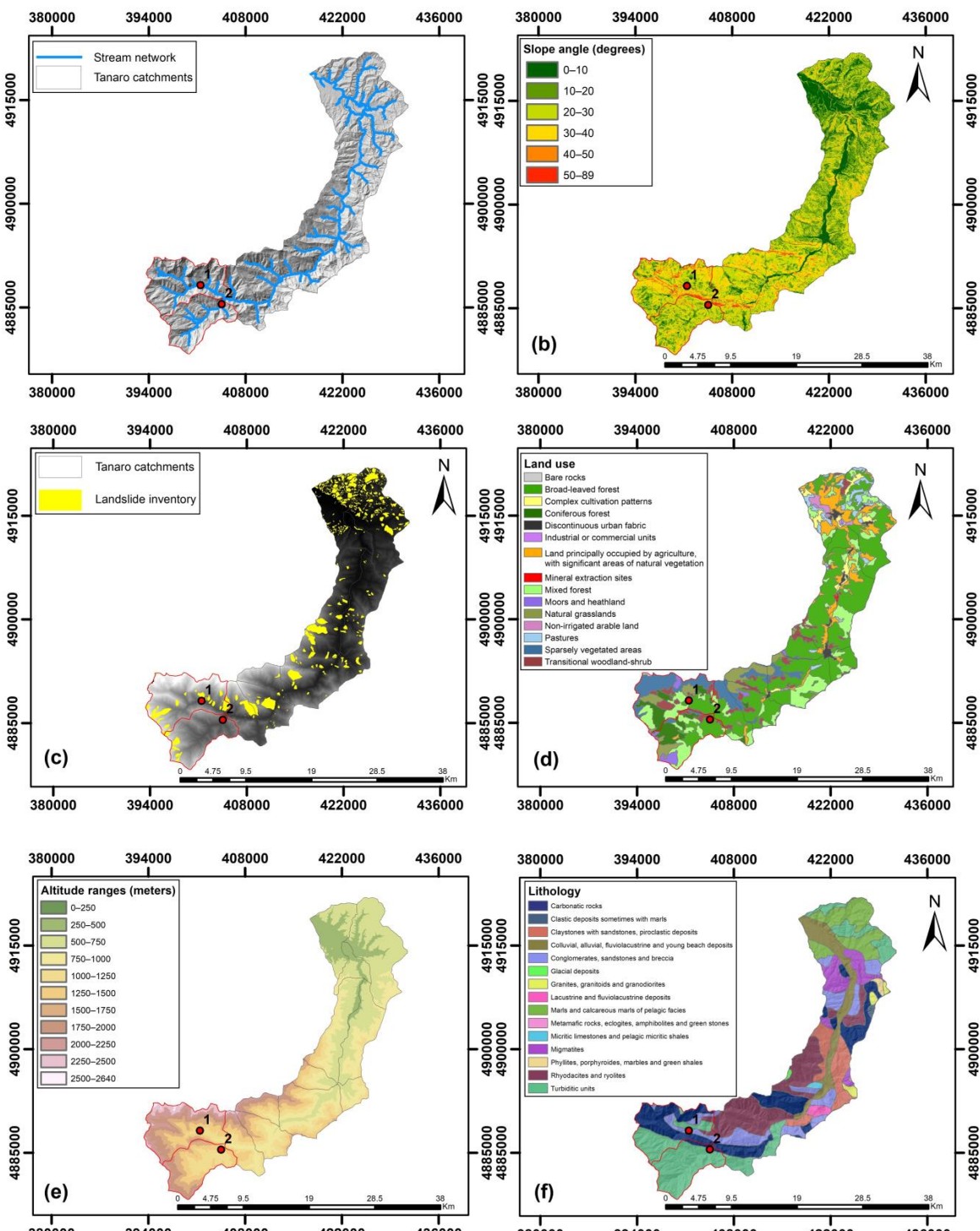

**Figure 3.** (**a**) Drainage network of the UTV, provided by [16,17]; (**b**) Slope; (**c**) 5 m resolution DTM of Liguria and Piemonte regions [18–20] and landslide distribution modified from [21]; (**d**) Land use map of Liguria Region and Piemonte Region [22,23]; (**e**) Elevation classification; (**f**) Lithological map modified from [24]. In red, the two largest secondary basins of the UTV are shown (Negrone River to the N and Tanarello River to the S), while the red dots indicate the position of the identified sites and their id number: 1 for the site on the Negrone River and 2 for the site on the Tanarello River.

## 3. Methods and Data

### 3.1. Preliminary Analysis and Site Selection

The preliminary analysis of the UTV consists of identifying the sub-basins and secondary catchment areas characterized by a surface area so that a significant fraction of the possible floods of the main river can be controlled by regulating the water flow of its outlet. This is more or less equal to finding catchments with an area of more than 40 km$^2$, i.e., 10% of the complete basin of the UTV.

Once the secondary basins have been selected, the choice of the position for the mitigation structures can take place. Four parameters must be considered:

− Interference with landslides;
− Interference with urban areas or infrastructures;
− Visibility of the site;
− Accessibility of the site.

The interference that the mitigation structure or the eventual lake may produce on landslides could provoke unwanted consequences, such as causing the activation of latent phenomena or the acceleration of already active ones, that could interact destructively with the structure itself. The interference with existing urban areas or infrastructures would be unacceptable. To better visualize this parameter, using the 5 m Digital Elevation Model (DTM) provided by Piemonte Region and Liguria Region [18,19], the extent of a lake produced by a 125 m-high dam has been modeled using QGis 3.16.14. The visibility of the site takes into account the visual impact of an unpleasant structure made of concrete. Lastly, the accessibility of the site aims to keep in mind the logistical needs of the erection of such a structure and its costs.

This qualitative selection allows for a quick evaluation of the study area, without sacrificing reasonable aspects that are usually of concern for both the inhabitants and the local or regional administrations. The data used for this purpose have been obtained from official sources of the Piemonte Region and Liguria Region, such as the already cited DTM [18–20], orthophotos [25,26], and landslide inventory [21].

### 3.2. Geomorphological Mapping

Once the sites had been identified using the qualitative method described in Section 3.1, a geomorphological survey was carried out in the study area using standard field methodologies and remotely sensed data obtained by ortho-photogrammetry [25,26].

This process aimed to evaluate the local characteristics of the site with more details than those reasonably achievable using available data produced by the regional authorities.

### 3.3. Basin Characteristics and Hydrological Analysis

A hydrological analysis has been carried out to assess the water flow rate both along the Tanaro River at five control sections (CS) and for the Tanarello and Negrone valleys. The positions of CS were chosen directly within or close to the five main urban areas of the UTV involved during the late October 2020 flood: Ormea, Garessio, Bagnasco, Nucetto and Ceva. These points, already shown in Figure 2, considered outlets, divide the drainage basin of the UTV into four sub-basins, while the last position (at the town of Ceva) marks the closing of the main basin here analyzed. For the Tanarello and Negrone valleys, of which the outlet corresponds to the identified sites, the approach is slightly different.

The methodology used to characterize from the the sub-basins from the hydrological point of view follows the standard approach commonly employed for this kind of study in Italy: to evaluate the water flow rate, the equation of Metodo Razionale (i.e., Rational Method) [27] was considered:

$$Q_{max} = C \, i \, A \, k \tag{1}$$

In Equation (1), $Q_{max}$ is the value of the maximum water flow rate in m$^3$/s expected at the considered position, C is an adimensional coefficient that takes in account the surface runoff fraction of rainfall, i is the maximum rainfall intensity expected in the selected area

expressed in mm/h, A is the area in km$^2$ of the catchment, k is an adimensional constant used to correlate the units of measure of the other parameters.

The surface runoff coefficient was derived from the land use map of the area provided by the Regional administrations of the Piemonte Region and Liguria Region [22,23]. A simplified approach was employed, and the land use maps were re-classified using a Geographic Information System (GIS) software and considering only five classes: anthropic areas, grassland and fields, vegetated areas, water bodies, and rock outcrops. For each of the five classes, a specific value of C was defined, using bibliographical sources as reference [28], to take into account the differences caused by the soil type associated with each of the five land use classes and the surface slope: this is a simple yet effective method, commonly used in Italy. These partial values have then been averaged using the area (in km$^2$) of each land use class as weight, obtaining thus a value representative of the entire drainage basin considered. This process has been repeated for each sub-basin defined within the main basin. It is worth noting that a single value was used for the C coefficient for the rock outcrops, despite different lithologies are present in the area. This approach was considered sufficient given the very small percentage of the area that the rock outcrops occupy (3.3%) within the Upper Tanaro Basin and its sub-basins, which renders any compositional difference between different types of rocks practically negligible for the evaluation of the global C coefficient.

The maximum rainfall intensity expected in a given area has been obtained using the intensity-duration-frequency curve equation:

$$i = a \, t^{n-1} \tag{2}$$

This power law expresses the value of maximum expected rainfall intensity (i, in mm/h) as a function of time (t, in h) and two parameters (a and n) of which n is adimensional while a is rainfall depth (in mm). These two parameters can be derived for a given recurrence period (Tr) from a statistically significant period of measurement of rainfall values, usually at least 30 years long, using Gumbel's method [29] and. The Regional Agency for Environmental Protection of Piemonte Region (ARPA Piemonte) already provides the values of the a and n parameters for the regional territory, through a free and public web interface [30]: this simplified approach is the standard for professionals in Northern Italy and was chosen for our study for this reason. The regional territory is divided into squares of side length equal to 250 m, for which the pa and n parameters are provided. For this study, a number of these cells, roughly amounting to a tenth of the area of each sub-basin, has been consulted, and the respective values averaged. The a and n values considered were those related to a 200 years recurrence period, which is the standard maximum recurrence time the Italian law requires for this kind of analysis [31]. The time variable in Equation (2) must be considered equal to the concentration time ($t_c$), i.e., the interval of time required for water to flow from the most remote point of the watershed to its outlet, to evaluate the maximum contribution from every point of the drainage basin. The $t_c$ (in h) has been estimated using Giandotti's formula [32], empirically derived for basins of the Italian peninsula with a surface greater than 10 km$^2$. The equation is as follows:

$$t_c = (4 \, A^{0.5} + 1.5 \, L)/[0.8 \, (H_a - H_o)^{0.5}] \tag{3}$$

where A is the area of the drainage basin (in km$^2$), L is the length of the main stream extended to the watershed divide (in km), $H_a$ is the average elevation of the drainage basin, and $H_o$ is the elevation of the outlet. All these parameters can be easily obtained using a GIS software: in this study, the drainage system of the UTV was obtained from official data of Piemonte Region and Liguria Region [16,17], while the extent of the drainage sub-basins and their elevation data were derived from the 5 m resolution DTM provided by both Piemonte Region and Liguria Region [18,19]. The metadata provided with these DTMs state a precision of about 0.50 m on average. Knowing the value of $t_c$ for each considered basin, the maximum expected rainfall intensity could be calculated. Lastly, since the value

of A of each drainage basin and sub-basin considered is already known, the maximum expected water flow rate for the 200-year recurrence period could be evaluated.

For the Tanarello and Negrone valleys, the values of i calculated for the sub-basins defined by the five CS were used, so that when referring to a specific catchment sub-basin, the intensity and duration of its expected rainfall are extended to the Tanarello Basin and Negrone Basin.

The $Q_{max}$ value calculated with (1) for the sub-basins of the Tanarello and Negrone valleys was then used to estimate, with acceptable approximation, the volume that the proposed dams have to accommodate. This volume is usually obtained by means of integration over the function that describes the relation between time and the water flow rate during a flood (also known as a hydrograph). This function has an initial part where the water flow rate grows with time up to a peak value (i.e., $Q_{max}$) correspondent to the time at which all the basin is contributing (i.e., the $t_c$ value); then the flow rate decreases until it returns to the initial value. The curve, in its simplest representation, has the shape of a triangle, thus the value of the flood volume can easily be calculated as its area. Once the water volume of the expected flood is known, using the DTM of the drainage basins, it is possible to evaluate the volume manageable by a dam of a given height.

Using a trial and error process, different heights for the dams located at the two identified positions along the Negrone and Tanarello rivers were used to extract the portion of DTM between the elevation of the base of the structure and its maximum height. Then, an algorithm available for the QGis 3.16.14 software was employed, called *raster surface volume*. The algorithm simply evaluates, for each cell of the portion of DTM previously extracted, the difference between the elevation of the surface defined by the maximum water level in the dam, by definition equal to the surface covered by the extracted DTM, and the elevation of a reference base level, equal to the elevation of the base of the dam. Using the spatial resolution of the DTM, the volume is calculated. For increasing values of height of the two dams, the correspondent volumes were evaluated, until a value large enough to surpass that of the expected flood was found. The two dams found this way are hence called optimal scenario (OS), even though, in order to propose a cost-effective design solution, different combinations of height for the two dams have been assumed and analyzed.

Since the dams are not intended to work by completely containing the flood but as retention basins, an outflowing water flow rate ($Q_{out}$) has been quantified: for the OS, this is roughly equivalent to 10% of the maximum expected water flow rate at the dam position. This value guarantees significant retention of the flood. For lower heights of the two dams, since their available volume cannot completely contain the flood, the outflow of water is required to maximize the efficiency of the flood mitigation structure. In this case, the outgoing water flow rate can be calculated following the definition of the retention coefficient ($\eta$) given by [33]:

$$Q_{out} = \eta \, Q_{in} = [1 - (V_{dam}/V_{flood})] \, Q_{in} \qquad (4)$$

Here, $Q_{in}$ is the ingoing water flow rates (in $m^3/s$), while $V_{dam}$ and $V_{flood}$ are the maximum volumetric capacity of the dam and the volume of the flood (in $m^3$). Since the value of $Q_{in}$ is equal to $Q_{max}$ calculated for the sub-basins of the Tanarello and Negrone valleys, and both the volumes $V_{dam}$ and $V_{flood}$ have already been quantified, the value of $Q_{out}$ can be calculated.

Finally, the value of $Q_{max}$ calculated in the five CS along the Tanaro River has been used to quantify the mitigation effect of the different design solutions proposed: this has been simply achieved by confronting the maximum expected water flow rate at each CS without and with the flood mitigation dams. The mitigated $Q_{max}$ (later referred to as $Q_{residual}$) value has been calculated by considering only the value of $Q_{out}$ as the contribution

of the mitigated drainage sub-basins of Tanarello and Negron rivers. The mitigation effect (M) of the dams proposed in this study is quantified as a percentage calculated as follows:

$$M\ (\%) = (Q_{max} - Q_{residual})/Q_{max} \tag{5}$$

## 4. Results

### 4.1. Chosen Sites

Considering the parameters described in Section 3.1, two sites have been identified: a first one along the Negrone River and a second one along the Tanarello River. This choice was forced by the fact that in the uppermost part of the Tanaro Basin, the areas of the sub-catchments appear to be quite small, with only a couple of exceptions. Of these, the most notable are the Negrone and Tanarello valleys: the first one covers a surface of approximately 49 km$^2$, while the second one has an area of 48 km$^2$; both are therefore acceptably wide since their covered area is more than 40 km$^2$ and, most notably, their combined area is more or less equal to 25% of the UTV basin surface.

Site 1 (Figures 3 and 4), along the Negrone River, is located in a position near the residential area of Viozene (a village in Ormea municipality). Still, due to the pronounced V-shaped valley incised by the Negrone River, the site is completely invisible from Viozene or from the road that passes through the valley: both are in fact located almost 200 m above the valley bottom. Therefore, it is easy to see how, even for the exaggerated 125 m-high dam, there would be no interference between the structure, the residential area of Viozene and the roads in its proximity. In this position, the Negrone Valley manifests a very well-defined V-shaped morphology, with steep to very steep valley sides. The site is more or less easily accessible thanks to a dirt road that follows the Negrone River. The site has also been selected because of the presence of bedrock outcrops in the surrounding area: this condition is necessary for the foundations of the eventual DRR structures, that would otherwise require extensive excavation. Moreover, this section of the Negrone Valley appears to be more stable, with no recorded major landslide phenomena occurring there. As can be seen in Figures 4 and 5, the portions of the Negrone valley upstream or downstream with respect to site 1 are either almost entirely inaccessible or heavily affected by landslides. The area around site 1 is not to be interpreted as entirely stable but simply as not involved in the massive landslides visible elsewhere.

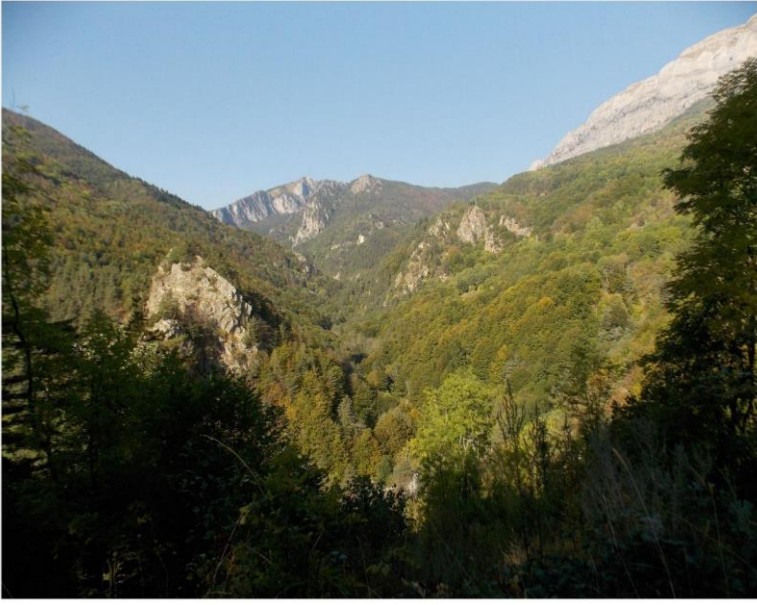

**Figure 4.** Picture of the upper portion of Negrone Valley, just beyond site 1.

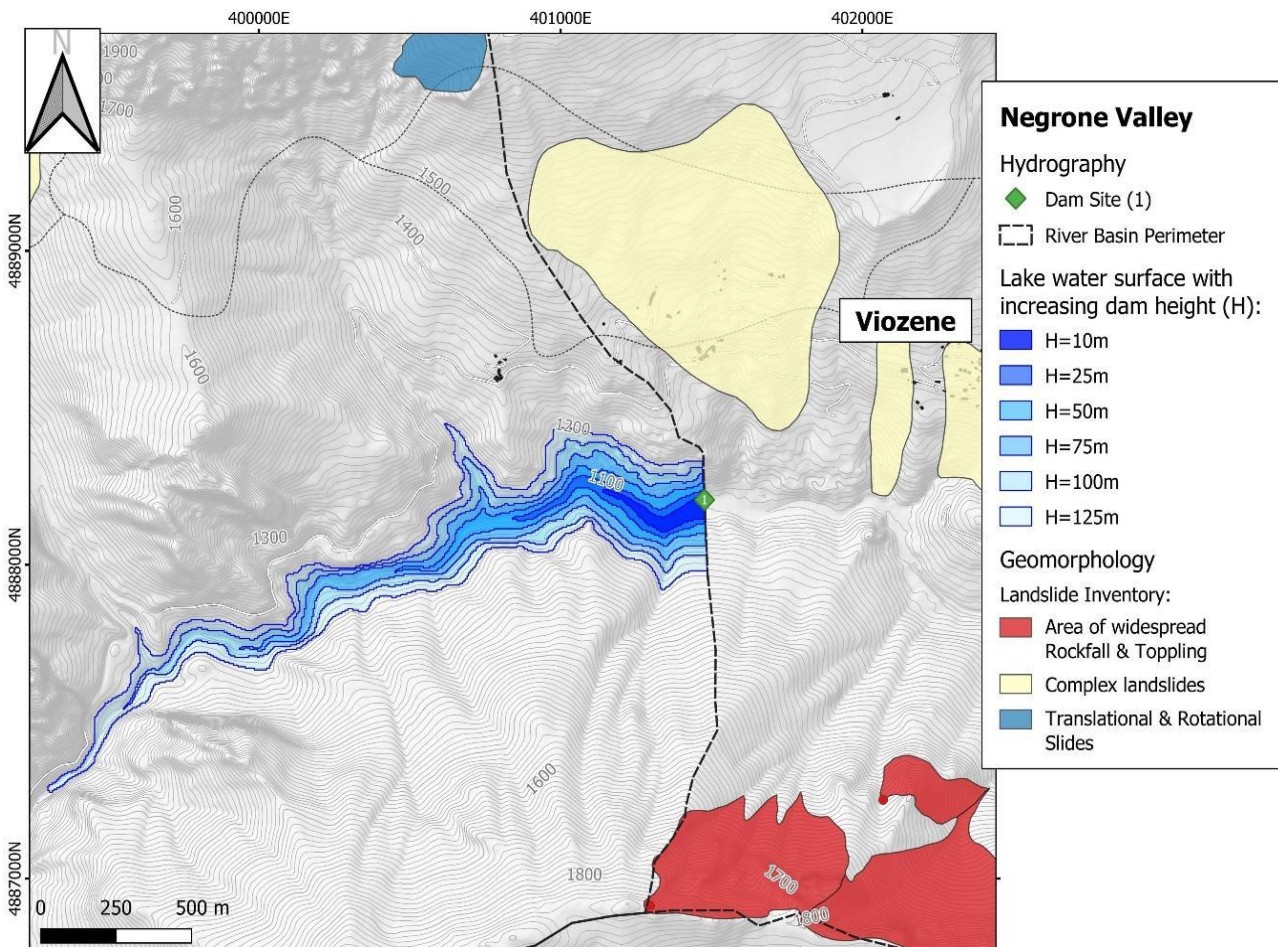

**Figure 5.** Surface of the lake produced by a dam 125 m-high located at site 1: note that there are no interactions with anthropic structures or known landslides according to [21].

Site 2 (Figures 6 and 7), along the Tanarello River, is located more or less 1 km upstream with regards to the confluence of the river with the Negrone River. The site is positioned just prior the gorge that characterizes the last section of the Tanarello River and precisely where a minor secondary stream flows into the Tanarello River. This position is too far from any anthropic structure or infrastructure to be easily seen without getting near to the site itself; moreover, the lower portion of the Tanarello Valley is completely uninhabited. The distribution of landslides recorded by [21] shows that the area is apparently the most stable, with only a few known landslides located along the stream or the valley sides. The bedrock in the area appears to be covered by a thin layer of colluvial deposits. The valley here has a rather wide valley bottom, mostly due to the presence of the secondary stream that flows into the Tanarello River. Lastly, the area is easily accessible thanks to a road that connects the lower part of the valley with the main road along the Tanaro River.

### 4.2. Geomorphological Map

The area around both sites 1 and 2 has been surveyed to verify the data provided by the regional administrations and expand it with more detail. Figures 8 and 9 show the data collected for the sites 1 and 2, respectively.

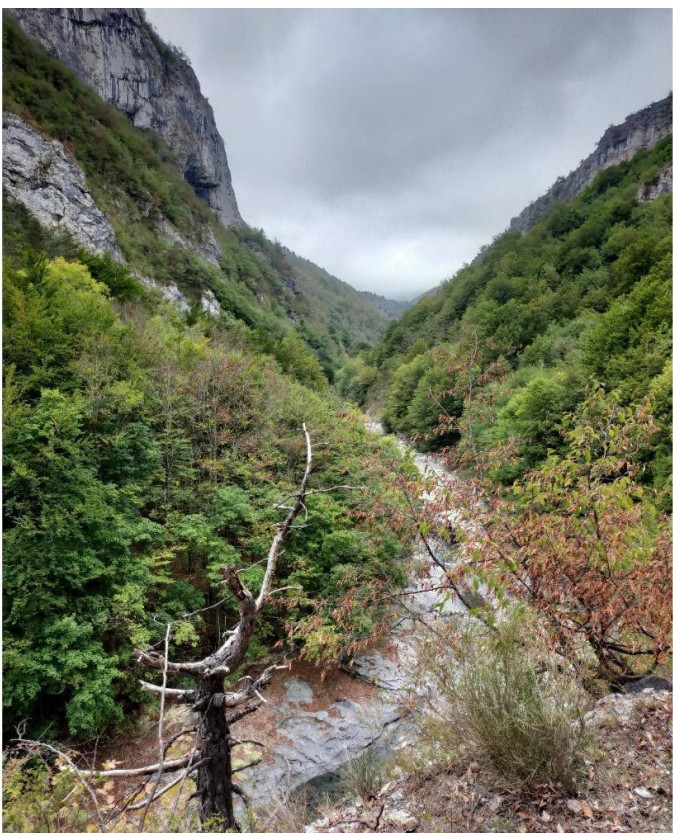

**Figure 6.** Picture of Tanarello Valley towards site 2.

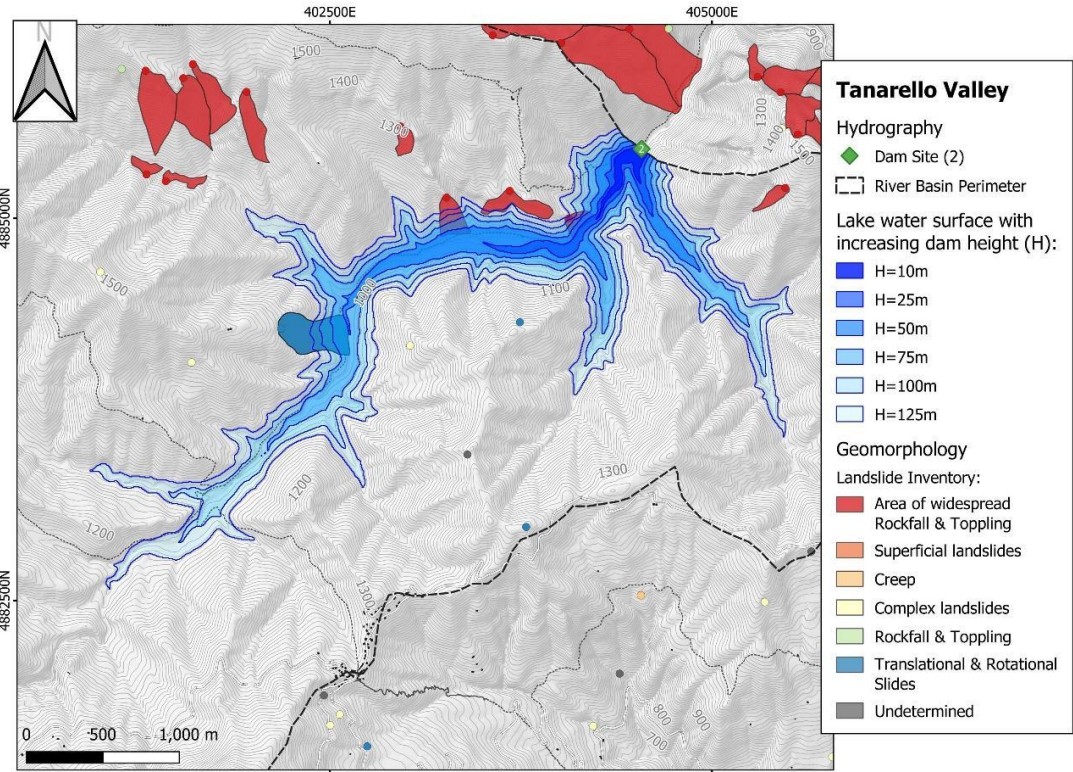

**Figure 7.** Surface of the lake produced by a dam 125 m-high located at site 2: note that there are no interactions with anthropic structures, but there are areas characterized by rockfall and slide processes along the river bed according to [21].

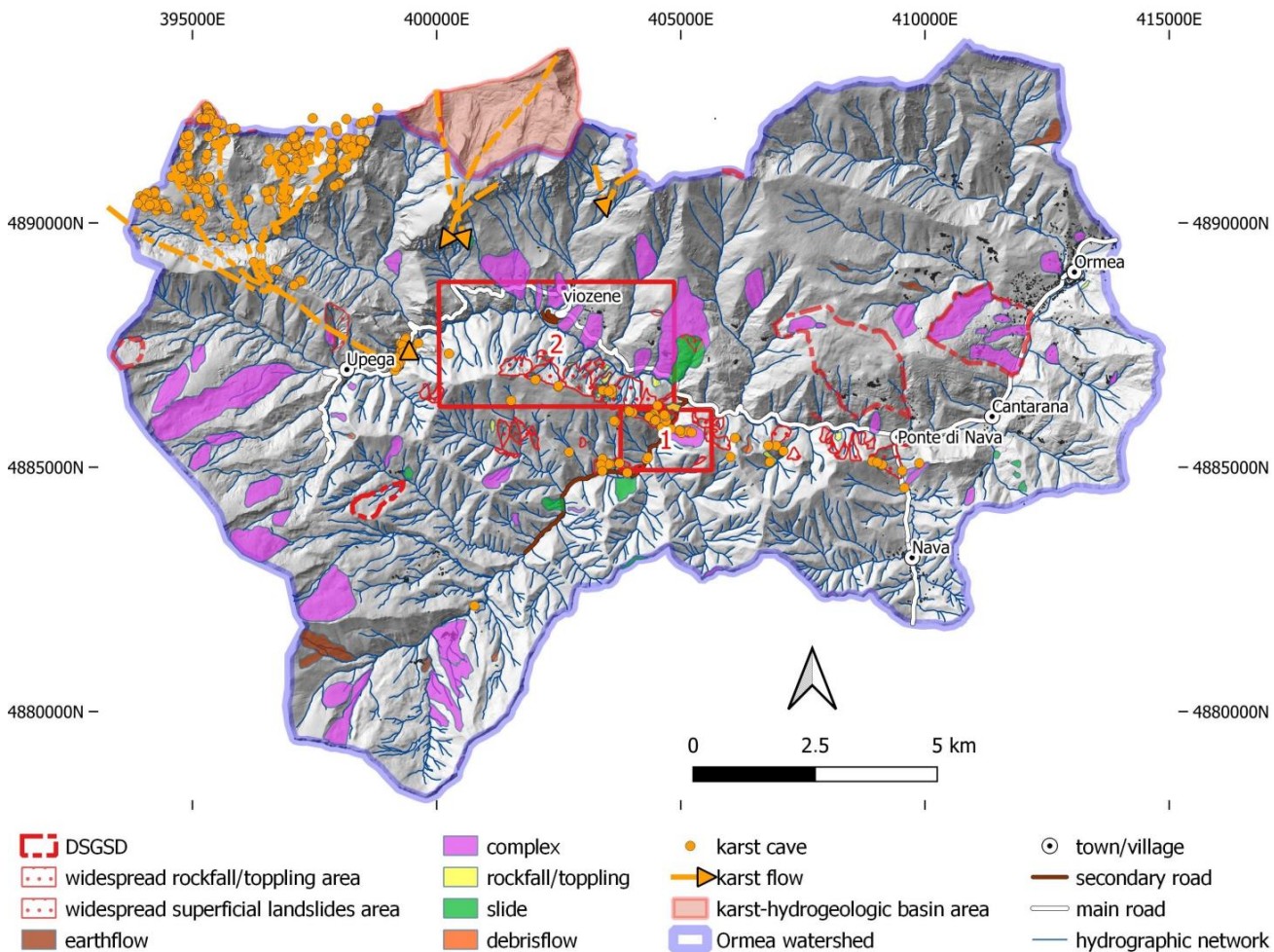

**Figure 8.** General geomorphological map of the uppermost portion of the UTV; the red boxes indicate the area surrounding the two sites, shown in the following Figure 9.

The area around site 1 appeared to be almost entirely stable: in reality, the data obtained by the geomorphological mapping carried out there showed that there are a certain number of landslides located along the river and the valley sides. These phenomena appear, though, to be related to instability of the shallow soil cover, usually 1–2 m thick, or to talus deposits located at the base of the extensive cliffs that mark the upper portions of the southern edge of Negrone Valley. It should also be noted how most of the secondary streams, which appear completely devoid of water, show signs of intense or very intense erosion, most likely the product of small debris flow-like phenomena. Of these, almost no deposits remain along the Negrone River. This new data does not change the evaluation carried out in the previous section nor its results, however: site 1 is still located in a very favorable location.

The area around site 2, unlike that around site 1, was described in Section 4.1 as having some landslides located along the Tanarello River or the valley sides. The field survey data show that those landslides are, in reality, of minor concern, since they are most likely the result of the interaction between sub-surface runoff and the 1–3 m thick soil cover. An aspect of key importance is the fact that the secondary stream that flows into the Tanarello River right at the chosen site appears to be much more geomorphologically active and tends to transport large quantities of sediments and debris into the Tanarello River.

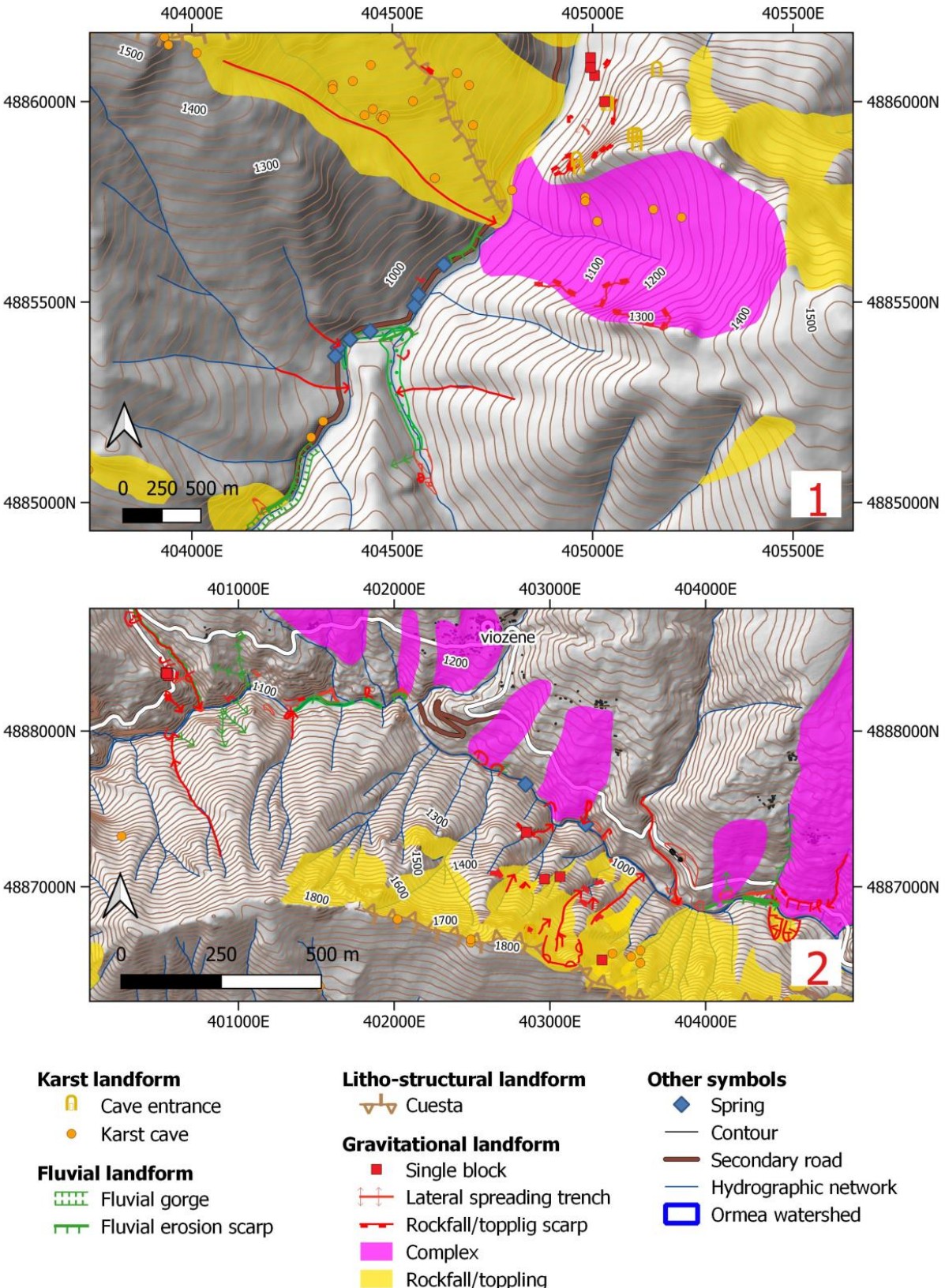

**Figure 9.** (**1**) Geomorphological map of the surroundings of site 1 along the Negrone River; (**2**) Geomorphological map of the surroundings of site 2 along the Tanarello River. Both maps are shown in the previous Figure 8 as red boxes.

### 4.3. Hydrological Analysis Results

As stated in Section 3.3, the first parameter needed in Equation (1) is the runoff coefficient (C), obtained for a chosen catchment from its land use map: for reference, in Figure 10, the re-classified land use map of the entire UTV is shown, with the relative area cover percentage for each of the five classes used. As it can be easily seen, the most widespread of the five land-use classes used corresponds to the areas occupied by vegetation of any kind. Table 1 shows the results of the evaluation of the C coefficient for the catchments of Tanarello River and Negrone River and the other five sub-basins.

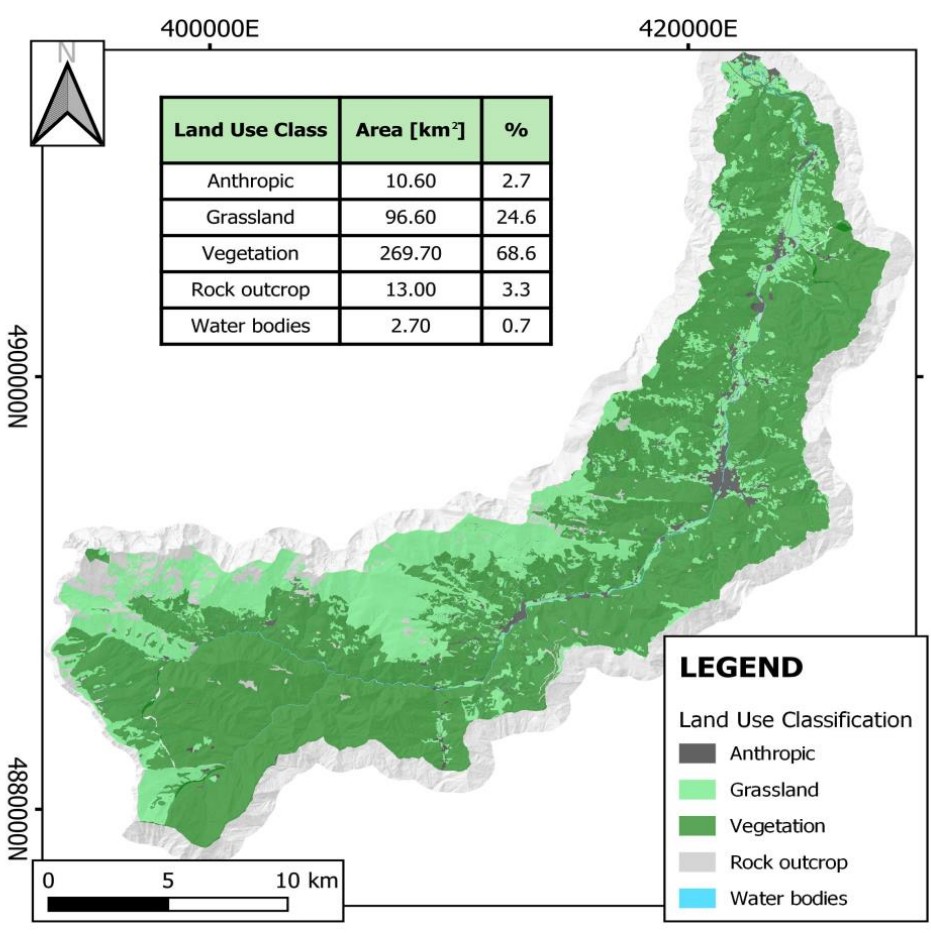

| Land Use Class | Area [km²] | % |
|---|---|---|
| Anthropic | 10.60 | 2.7 |
| Grassland | 96.60 | 24.6 |
| Vegetation | 269.70 | 68.6 |
| Rock outcrop | 13.00 | 3.3 |
| Water bodies | 2.70 | 0.7 |

**LEGEND**

Land Use Classification
- Anthropic
- Grassland
- Vegetation
- Rock outcrop
- Water bodies

**Figure 10.** Re-classified land use map for the complete UTV basin with the percentage of area associated with each of the five classes considered.

**Table 1.** Evaluation of the C coefficient for each sub-basin of the UTV and Tanarello and Negrone valleys.

| Land Use Class | Ceva [km²] | Nucetto [km²] | Bagnasco [km²] | Garessio [km²] | Ormea [km²] | Negrone V. [km²] | Tanarello V. [km²] | Land Use Class C |
|---|---|---|---|---|---|---|---|---|
| Anthropic | 10.60 | 9.56 | 8.82 | 5.06 | 2.29 | 0.17 | 0.22 | 0.90 |
| Grassland | 96.59 | 92.60 | 87.57 | 75.94 | 58.67 | 16.74 | 5.12 | 0.62 |
| Vegetation | 269.71 | 258.51 | 246.42 | 184.23 | 121.08 | 23.03 | 42.55 | 0.36 |
| Water | 2.74 | 2.48 | 2.35 | 1.81 | 1.17 | 0.30 | 0.13 | 1.00 |
| Rocks | 2.74 | 12.89 | 12.85 | 12.49 | 11.16 | 7.67 | 0.44 | 0.70 |
| **C value for each basin** | 0.45 | 0.45 | 0.45 | 0.46 | 0.47 | 0.51 | 0.39 | |

The results show an increase in the C coefficient values as the area of the sub-basin decreases and its position moves westward toward the uppermost portion of the basin: this is due to the decreasing presence of vegetated areas and the broader distribution of grassland. The significant difference between the C coefficient for Tanarello Valley and Negrone Valley has a similar explanation. It should be noted how low the percentage of area covered by the rock outcrop class is, equal to 3.3% for the whole basin of the UTV. Table 2 shows the results of Equation (3) for each sub-basin, while Table 3 summarizes the results of the rainfall intensity evaluation using Equation (2).

**Table 2.** Parameters required by (3) for the evaluation of the concentration time ($t_c$) and results for each of the five sub-basins of the UTV and for Negrone and Tanarello valleys.

| Basins | A | L | $H_a$ | $H_o$ | $t_c$ |
|---|---|---|---|---|---|
| | [km$^2$] | [km] | [m] | [m] | [h] |
| Ceva | 393 | 64 | 1196 | 372 | 7.66 |
| Nucetto | 376 | 56 | 1224 | 446 | 7.24 |
| Bagnasco | 358 | 52 | 1253 | 470 | 6.85 |
| Garessio | 280 | 38 | 1380 | 572 | 5.44 |
| Ormea | 195 | 26 | 1518 | 715 | 4.17 |
| Negrone V. | 49 | 10 | 1853 | 1066 | 1.91 |
| Tanarello V. | 48 | 11 | 1467 | 941 | 2.41 |

**Table 3.** Evaluation of the maximum expected rainfall intensity (i) according to Equation (2) for a recurrence period of 200 years, calculated for the five sub-basin of the UTV.

| Basins | $t_c$ | a | n | i |
|---|---|---|---|---|
| | [h] | [mm] | [/] | [mm/h] |
| Ceva | 7.66 | 65.099 | 0.472 | 22.26 |
| Nucetto | 7.24 | 65.658 | 0.483 | 23.61 |
| Bagnasco | 6.85 | 65.727 | 0.497 | 24.95 |
| Garessio | 5.44 | 65.646 | 0.508 | 28.50 |
| Ormea | 4.17 | 64.328 | 0.510 | 31.91 |

Since the k parameter of Equation (1) is a constant, the last variable to quantify is the area of the catchment basin (A), easily obtainable using QGis software. Table 4 summarizes the evaluation of the maximum water flow rate ($Q_{max}$) expected at the considered positions. It should be noted that the lowering of the $Q_{max}$ values for the larger sub-basins, identified by the control position at Nucetto and Ormea, is to be related to the position of the CS themselves: the towns of Nucetto and Ormea are located in the terminal portion of the UTV, where the subsequent Tanaro Plain begins. Therefore, the rainfall phenomena occurring there are not constrained by the mountains of the upper portion of the Tanaro Basin. Figure 11 shows the position of the CS and their respective sub-basins.

**Table 4.** Parameters required by Equation (1) and the maximum expected water flow rate ($Q_{max}$) for a 200-year recurrence period at each CS along the Tanaro River.

| Basins | C | i | A | k | $Q_{max}$ |
|---|---|---|---|---|---|
| | | [mm/h] | [km$^2$] | | [m$^3$/s] |
| Ceva | 0.45 | 22.26 | 393 | 0.278 | 1094 |
| Nucetto | 0.45 | 23.61 | 376 | 0.278 | 1110 |
| Bagnasco | 0.45 | 24.95 | 358 | 0.278 | 1117 |
| Garessio | 0.46 | 28.50 | 280 | 0.278 | 1020 |
| Ormea | 0.47 | 31.91 | 195 | 0.278 | 813 |

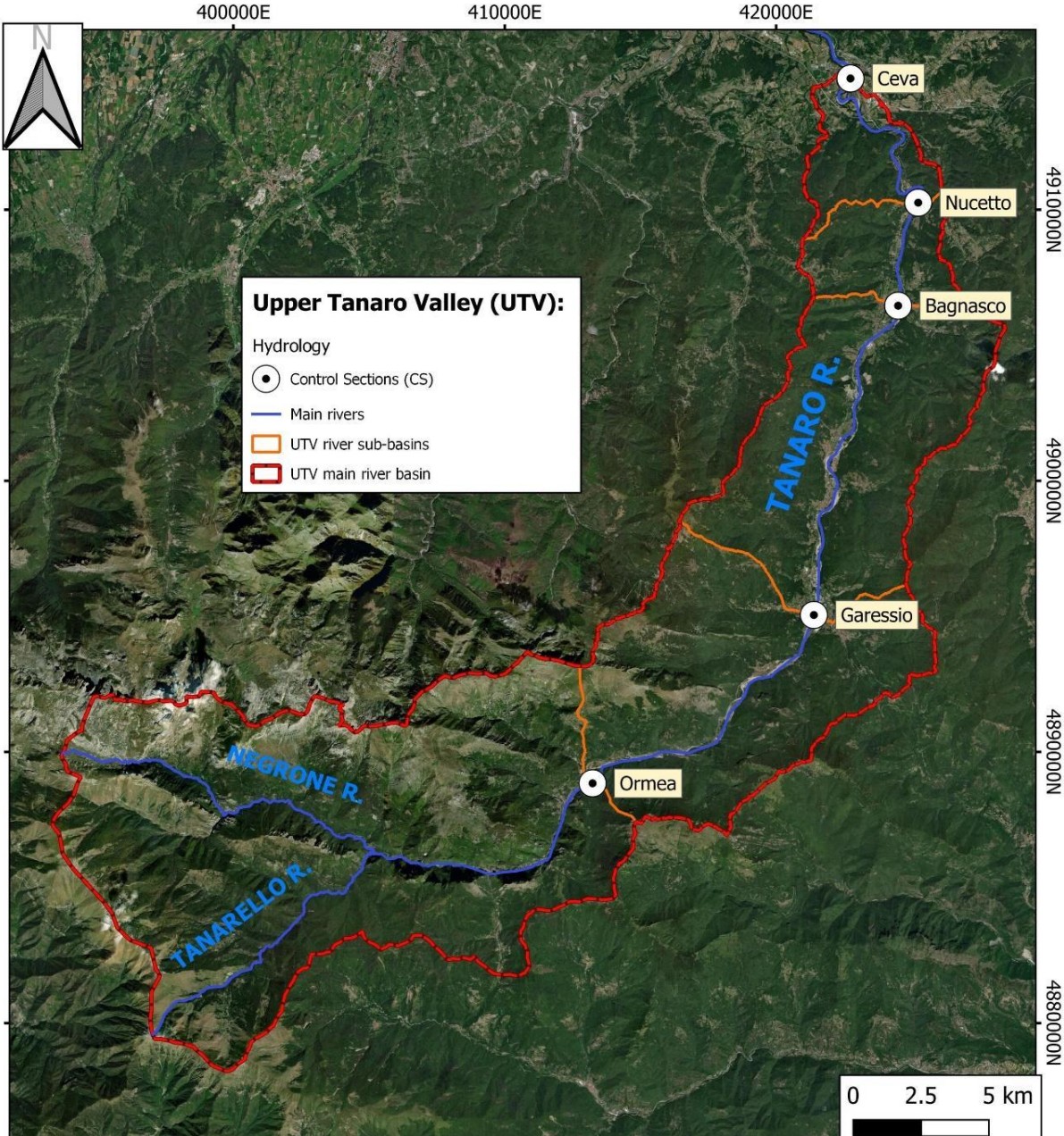

**Figure 11.** The five control sections located in the UTV main urban areas and the sub-basins they define.

By calculating the area of a triangle representing the ideal flood hydrograph, the equivalent maximum expected volume of floodwater for the position of the two dams along the Negrone and Tanarello rivers is evaluated, considering the expected rainfall intensity assessed at each of the five control positions along the Tanaro River: results are shown in Table 5. Then, the height of the dams for both selected positions able to contain the flood water volume is determined: for the dam positioned on the Negrone River, its optimal height is equal to 63 m, while for the dam on the Tanarello River it is 34 m. This is the OS, which allows only 10% of the maximum entering water flow rate and represents the best possible risk mitigation scenario: the outgoing water flow rate for the dam on the Negrone River is equal to 21 m$^3$/s, and to 17 m$^3$/s for the dam on the Tanarello River. By comparing the values of the water flow rate at the five control positions with and without the dams in this configuration, the mitigation effect can be appreciated (Table 6).

**Table 5.** Evaluation of the maximum expected flood volume at the dam located along the Negrone River and the Tanarello River, obtained using the rainfall intensity values calculated at each of the five control positions along the Tanaro River.

| Control Positions | Ceva | | Nucetto | | Bagnasco | | Garessio | | Ormea | |
|---|---|---|---|---|---|---|---|---|---|---|
| **Negrone Valley** | $Q_{max}$ $[m^3/s]$ | $V_{flood}$ $[m^3]$ | $Q_{max}$ $[m^3/s]$ | $V_{flood}$ $[m^3]$ | $Q_{max}$ $[m^3/s]$ | $V_{flood}$ $[m^3]$ | $Q_{max}$ $[m^3/s]$ | $V_{flood}$ $[m^3]$ | $Q_{max}$ $[m^3/s]$ | $V_{flood}$ $[m^3]$ |
| $t_c = 1.91$ h | 154.1 | $4.2 \times 10^6$ | 163.4 | $4.3 \times 10^6$ | 172.7 | $4.3 \times 10^6$ | 197.3 | $3.9 \times 10^6$ | 220.9 | $3.3 \times 10^6$ |
| **Tanarello Valley** | $Q_{max}$ $[m^3/s]$ | $V_{flood}$ $[m^3]$ | $Q_{max}$ $[m^3/s]$ | $V_{flood}$ $[m^3]$ | $Q_{max}$ $[m^3/s]$ | $V_{flood}$ $[m^3]$ | $Q_{max}$ $[m^3/s]$ | $V_{flood}$ $[m^3]$ | $Q_{max}$ $[m^3/s]$ | $V_{flood}$ $[m^3]$ |
| $t_c = 2.41$ h | 116.8 | $3.2 \times 10^6$ | 123.9 | $3.2 \times 10^6$ | 130.9 | $3.2 \times 10^6$ | 149.6 | $2.9 \times 10^6$ | 167.5 | $2.5 \times 10^6$ |

**Table 6.** The OS for the two dams proposed for the Negrone and Tanarello rivers and their mitigation effect (M) expressed as a percentage; the $Q_{out}$ values for the Negrone and Tanarello dams are 21 m$^3$/s and 17 m$^3$/s, respectively (thus, a combined 38 m$^3$/s).

| CS | $Q_{max}$ | $Q_{out}$ | $Q_{residual}$ | Mitigation (M) |
|---|---|---|---|---|
| | $[m^3/s]$ | $[m^3/s]$ | $[m^3/s]$ | % |
| Ormea | 813 | 38 | 458 | 44 |
| Garessio | 1020 | 38 | 709 | 30 |
| Bagnasco | 1117 | 38 | 858 | 23 |
| Nucetto | 1110 | 38 | 869 | 22 |
| Ceva | 1094 | 38 | 870 | 20 |

Since the height of the two dams in the OS is considerable, particularly for the one proposed on the Negrone River, the cost of realizing such structures is very high: therefore, we have also analyzed the effect of other configurations. First of all, the impact of each of the two "optimal" dam configurations is considered, then the mitigation produced by each of the two dams if their height was only 30 m, and lastly, their combined mitigation effect. These five models are assessed in Tables 7–11.

**Table 7.** The mitigation effect of the single Negrone River dam as for its OS; the outgoing flow rate is equal to 21 m$^3$/s.

| CS | $Q_{max}$ | $Q_{out}$ | $Q_{residual}$ | Mitigation (M) |
|---|---|---|---|---|
| | $[m^3/s]$ | $[m^3/s]$ | $[m^3/s]$ | % |
| Ormea | 813 | 21 | 610 | 25 |
| Garessio | 1020 | 21 | 844 | 17 |
| Bagnasco | 1117 | 21 | 974 | 13 |
| Nucetto | 1110 | 21 | 977 | 12 |
| Ceva | 1094 | 21 | 972 | 11 |

**Table 8.** The mitigation effect of the single Tanarello River dam as for its OS; the outgoing flow rate is equal to 17 m$^3$/s.

| CS | $Q_{max}$ | $Q_{out}$ | $Q_{residual}$ | Mitigation (M) |
|---|---|---|---|---|
| | $[m^3/s]$ | $[m^3/s]$ | $[m^3/s]$ | % |
| Ormea | 813 | 17 | 658 | 19 |
| Garessio | 1020 | 17 | 886 | 13 |
| Bagnasco | 1117 | 17 | 1010 | 10 |
| Nucetto | 1110 | 17 | 1011 | 9 |
| Ceva | 1094 | 17 | 1094 | 8 |

**Table 9.** The mitigation effect of the single Negrone River dam if it had a height of 30 m; the outgoing flow rate, calculated to optimize the efficiency of the retention dam using Equation (4), is equal to 171 m$^3$/s.

| CS | $Q_{max}$ | $Q_{out}$ | $Q_{residual}$ | Mitigation (M) |
|---|---|---|---|---|
|  | [m$^3$/s] | [m$^3$/s] | [m$^3$/s] | % |
| Ormea | 813 | 171 | 760 | 7 |
| Garessio | 1020 | 171 | 994 | 3 |
| Bagnasco | 1117 | 171 | 1117 | 0 |
| Nucetto | 1110 | 171 | 1110 | 0 |
| Ceva | 1094 | 171 | 1094 | 0 |

**Table 10.** The mitigation effect of the single Tanarello River dam if it had a height of 30 m; the outgoing flow rate, calculated to optimize the efficiency of the retention dam using Equation (4), is equal to 37 m$^3$/s.

| CS | $Q_{max}$ | $Q_{out}$ | $Q_{residual}$ | Mitigation (M) |
|---|---|---|---|---|
|  | [m$^3$/s] | [m$^3$/s] | [m$^3$/s] | % |
| Ormea | 813 | 37 | 677 | 17 |
| Garessio | 1020 | 37 | 905 | 11 |
| Bagnasco | 1117 | 37 | 1030 | 8 |
| Nucetto | 1110 | 37 | 1030 | 7 |
| Ceva | 1094 | 37 | 1023 | 7 |

**Table 11.** The combined mitigation effect of the Negrone and Tanarello Rivers dams if they had a height of 30 m; the combined outgoing flow rate is equal to 208 m$^3$/s, but is valid only for the first two control positions; elsewhere only the 37 m$^3$/s value of the Tanarello River is shown.

| CS | $Q_{max}$ | $Q_{out}$ | $Q_{residual}$ | Mitigation (M) |
|---|---|---|---|---|
|  | [m$^3$/s] | [m$^3$/s] | [m$^3$/s] | % |
| Ormea | 813 | 208 | 627 | 23 |
| Garessio | 1020 | 208 | 879 | 14 |
| Bagnasco | 1117 | 37 | 1030 | 8 |
| Nucetto | 1110 | 37 | 1030 | 7 |
| Ceva | 1094 | 37 | 1023 | 7 |

It should be noted that for the model describing a single 30 m high dam on the Negrone River, the M (%) value of zero corresponds to the fact that the outgoing flow rate calculated using Equation (4) is higher than the ingoing water flow rate. This simply means that in order to accommodate the flood volume, this dam cannot retain anything: this is equivalent to saying that the dam is not working in any way, letting all the flood water pass and therefore producing no effects downstream. Furthermore, for the last model with two 30 m high dams along both Negrone and Tanarello rivers, the effect of the first dam (i.e., the one on Negrone River) is accounted only for the first two CS, due to its limited volumetric capacity: this means that for the other control positions the presence of this dam is neglectable and it is treated as if it was not even present; for the evaluation of $Q_{residual}$ on the last three control positions the contribution in terms of surface runoff and water flow of Negrone Valley is equal to 100% (i.e., the non-mitigated water flow). This is why, in Table 11 the $Q_{out}$ value changes from 208 m$^3$/s (combined effects of both dams) to 37 m$^3$/s (effects of the Tanarello River dam).

Since the goal was to identify the best cost-effective solution for flood risk mitigation in the UTV, by confronting the mitigation factors expressed as a percentage in Tables 7–11, it can be easily seen that the OS is the best choice in terms of mitigation, but disregards completely the economical aspect of the problem. It is also easy to see that of the two drainage sub-basins of Negrone and Tanarello, the first one is more important in terms of

volume of water expected for a flood with a 200-year recurrence time, but is also the most disadvantageous to operationally, since the optimal height of its dam is more than 60 m, which is unreasonable. It should also be noted that for lower heights, the effectiveness of the dam on the Negrone River decreases substantially; Tanarello Valley is much more promising, since the optimal height of its dam is 34 m.

## 5. Discussion

As the data reported in Section 4.3 suggest, the maximum percentage of mitigation achievable for the UTV, i.e., the data described for the Ceva CS, is approximately 25%, which is relatable to the fact that the area of the Tanarello and Negrone catchments combined amount to approximately a quarter of the UTV basin: this percentage is reduced by the fact that even for the OS, the two dams are always associated with an outgoing water flow rate, assumed equal to 10% of the ingoing flow rate. Thus, the value of almost 25% is an ideal upper limit, achievable only if the two sub-basins are wholly isolated from the rest of the basin. This is equivalent to considering the dams proposed as if they were to work as proper dams, retaining 100% of the water during a flood. For the other smaller sub-basins identified by the remaining four CS the mitigation effect is higher, due to the reduction in their watershed area, up to an ideal maximum of 50% at the uppermost CS located right into the town of Ormea.

Another aspect of crucial importance is the significant difference in cost-effectiveness between the Negrone River dam and the Tanarello River one. This aspect is related to the morphological configuration of the two valleys: the Negrone Valley is more deeply cut into the rocks of the area and is characterized by steeper sides; therefore, to achieve significant volumetric capacity, its dam requires a considerable height. This, considering the parameters used to evaluate and choose the sites for the dams, is perfectly acceptable: the site along the Negrone River is easily accessible, difficult or near impossible to see from the local residential areas or infrastructures, and does not interfere with them, nor does the dam produce a lake that interacts with known landslides or potential unstable locations. The problem arises when taking into account that the extreme height of the proposed dam determines costs that are too high. As the alternative 30 m high dam model shows, the volumetric capacity of the structure and its mitigation effect, decreases quickly with its height, making de facto the dam useless if not in the OS. This is also proven by the fact that with a height of 30 m, the Negrone River dam would only produce notable mitigation effects on the $Q_{max}$ values of the uppermost CS at Ormea and Garessio, and even in this case, the consequences would be minimal. This quick cost-effectiveness analysis suggests that the Negrone River site cannot be the leading choice for a DRR structure, or at least not the primary option. This analysis also shows that the configuration composed of two 30 m high dams is useless since the one on Negrone River would provide very limited to no mitigation.

On the other hand, the Tanarello Valley offers better conditions: the morphology of the valley is more favorable, with a wider bottom and less steep sides. The chosen site is still easy to access, invisible from the nearest residential areas or infrastructures, and the eventual lake produced by the dam does not interfere with unstable sectors of the area or infrastructures and other buildings, since the lower portion of the valley is uninhabited. From a cost-effectiveness point of view, this site is the most valuable given the fact that the height of the proposed dam, even for the OS, is quite reasonable (34 m), and the difference in terms of mitigation effect if the lower 30 m high dam model is considered is small (approximately 2% lower). It should also be mentioned that the orientation of the Tanarello Valley is more closely similar to the direction from which the storms and other intense rainfall events tend to come, as previously shown in [2]. All considered the Tanarello River site seems to be the most reasonable and cost-effective choice, or at least should be considered as the primary choice if the will to operate on both valleys by the local or regional administrations is shown.

## 6. Conclusions

The public data provided by the administrations of Piemonte Region and Liguria Region and additional data obtained by field surveys in the study area were used to perform a preliminary site selection for the risk mitigation structures and evaluate their efficiency.

The results show that of the two initial chosen sites, the most promising is the second one, along the Tanarello River: the morphology of the valley allows for a dam with high volumetric capacity even for relatively low height structure, as shown by the models proposed in this article. The first site, although initially very promising, due to its V-shaped morphology and steep valley sides, would require a dam almost twice as high as the one on the Tanarello River to achieve significant flood mitigation.

Therefore, this study suggests that the interest of local and regional administration in building structures such as those here considered should be focused on the Tanarello Valley, more specifically on site 2, because of its higher ratio between costs and benefits in terms of flood mitigation.

This study highlights the importance of an integrated approach based on geomorphological, geo-hydrological, and hydraulic evaluations when dealing with the choice of a flood risk mitigation strategy, even when considering the economic aspect of the problem. In fact, the simple task of choosing the sites within Tanarello and Negrone valleys, cannot be carried out correctly without complementing the hydrological data with geomorphological information regarding landslides distribution. Even the economic aspect is deeply influenced by the geomorphological and geohydrological features of the sites: in fact, it is the significant difference between the geomorphology of the Negrone Valley and the geomorphology of the Tanarello Valley that explains the need for an excessively high dam at site 1, therefore lowering its cost-effectiveness.

In the end, though, the method proposed and the results we have shown for the UTV basin are always to be intended as a preliminary study. In the circumstance that one or both of the sites here identified are actually chosen for the construction of a DRR structure, further and more detailed studies are absolutely required, along with hydraulic numerical simulations at the scale of the entire basin.

**Author Contributions:** Conceptualization, B.T. and M.B.; Data curation, B.T., M.L. and V.B.T.; Formal analysis, B.T.; Funding acquisition, G.U.; Investigation, B.T., M.L. and M.B.; Methodology, B.T.; Software, B.T., M.L., V.B.T. and M.B.; Validation, B.T.; Writing—original draft, B.T., M.L. and V.B.T.; Writing—review and editing, B.T., M.L., V.B.T., M.B. and G.U. All authors have read and agreed to the published version of the manuscript.

**Funding:** This research was funded by Fondazione CRC-Sessione Erogativa Generale 2021—Contributo Straordinario Finalizzato per lo "Studio preliminare alla realizzazione di sbarramenti nella zona a monte di Ormea per la mitigazione del rischio geo-idrologico nell'Alta Val Tanaro".

**Data Availability Statement:** The data presented in this study are available on request from the corresponding author. The data are not publicly available due to the fact that they are part of a project which have not yet been concluded and, therefore, not published.

**Acknowledgments:** The authors would like to thank L. Masciocco, A. M. Ferrero and G. Fubelli of the Department of Earth Sciences of University of Turin, who have prompted this study and provided assistance during its development. The authors are greatful to Laurie J. Kurilla for her linguistical support.

**Conflicts of Interest:** The authors declare no conflict of interest.

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
