# Peer review of "Proposal for Flood Risk Mitigation in the Upper Tanaro Valley (Western Alps—North-Western Italy)"

_geosciences, doi:10.3390/geosciences12070260_

Round 1
Reviewer 1 Report
The MS raises an important question: how to mitigate disastrous floods. However, in the present form the manuscript is badly written, not well organised, and with low international interest. Without any actual hydrological data, it is difficult to evaluate the correctness of the calculation.
Abstract
Please highlight your results in greater depth.
Introduction:
Please, define flood risk.
You should start with general statements instead of jumping into the Italian situation.
The introduction is not a review of flood risk assessment methods, nor an introduction on the causes of increased flood hazard. No open scientific questions are raised, just a case study is prepared.
L. 35: the reference at the beginning of the sub sentence is very strange: “although in [1] the flood” similar is “[3]” – and probably at other places as well… Please check, and create well-organized and shorter! Sentences, like: “According to Lastoria et al. [1] the flood activity of the Tanaro River is limited or very limited…”
L. 42: Nobody knows where is Ceva (you write for an international journal!)
L. 71: What is DDR?
Study area
The geology is unnecessarily described in such a great depth, but nothing is written on precipitation, infiltration run-off, channel parameters, valley density, flow regime, discharge, transported sediments etc. which are the most important elements of flood development.
L. 105- 128: The description of the geology of the area is too detailed, it is not necessary. The abbreviations of the units are not needed, as they are not mentioned any more.
L. 161-171: The description of the mass movements is too detailed. Please, shorten it and be more focused on the actual density, activity.
Methods
L. 176: “wide enough” wide area? What do you mean by enough? It is science, you have to express it by numbers/proportions.
L. 186-199: What was the database of the mentioned parameters? What was their spatial resolution? In the present form the study is not replicable, as you did not clearly describe your methods.
Fig.2: the font size is too small to read. The title of the figure is strange with the lots of “from”. Please, check my previous suggestion + check the Author Guide of the Journal.
L. 215: “five control sections (CS)“ where are they? Please, indicate on a map.
L. 218: Please indicate these villages on a map, nobody knows where are they…
L. 232: “surface runoff coefficient was derived from the land use map” so you did not consider the infiltration (dependent on also soil type and geology)? Please, clarify.
You calculated discharge, rainfall intensity, but as far as you mention no actual measured data, it is impossible to evaluate the accuracy of your method. It is a key point!!
Results
L. 327: Indicate on a general map, because i.e. based on Fig4 it is impossible to locate it in reality.
L. 344: “because of the presence of bedrock outcrops” why is it favorable?
L. 354: “The area appears to be stable, with only a few known landslides located along the river stream or the valley sides, which are not particularly steep” I think is a very weak approach, as even 1-2 landslides can cause fatal problems (see Vaiont Dam).
I do not see where is the site 1 and 2 on Fig. 7-8. Somehow the manuscript is totally confusing in a spatial point of view. The applied order of the sites is very strange: introduction of the sites, than the entire area, than two more area (I have no idea what is their connection to the selected sites).
L. 464: What is the accuracy of the calculated parameters? How do these data fit to the actually measured ones?
In the tables, please, use the same font type.The English of the manuscript is quite poor. An intensive English check of the text is needed, as there are strange sentences… (L.15: “the Tanaro River basin: in the last decades, the 16 Tanaro river” or “public available data”, or “Gis”, etc.)
Reviewer 2 Report
The manuscript entitled “Proposal for the flood risk mitigation in the Upper Tanaro Valley (Western Alps – Northwestern Italy)” is of a very interesting topic. The authors attempt to identify suitable sites for the construction of dams having the function of retention basins, aiming to mitigate the flood risk in the river basin of the Upper Tanaro Valley. They qualitatively identified the suitable locations using public available data and field-work, while they also applied several models to evaluate the hydrological characteristics of the study area and to assess the efficiency of each site in terms of flood water flow rate reduction. The topic of the paper is totally within the scopes of the journal “Geosciences”.
The paper constitutes an integrated geomorphologic, geohydrologic approach using also hydraulic evaluations which is quite important as a tool for flood risk mitigation plans. It is a great work which provides a proposal for similar integrated approaches to solve flood problems. The manuscript is well structured and very explanatory! The maps of the paper are great! I am not a native speaker but in my opinion the paper reads quite well!
Here are some comments and suggestions of minor importance that in my opinion could improve the final version.
lines 5-9: I think that the affiliation should appear once having at the end of the affiliation the emails of the authors.
line 98: I propose to change the title of the section “Setting” to “Study Area”.
line 173: I also propose to change the title of the “Methodologies” section to “Methods and data”.
page 6: In my opinion Figure 2 is part of the results and should not be in the “Methodology” setion.
lines 217-218: The sites/urban areas of Ormea, Garessio, Bagnasco, Nucetto and Ceva should appear in a map. Maybe the authors should add an inset map in Figure 1 with all the names of the sites mentioned within the text.
I think that a flow diagram showing the main methodological steps followed should b appreciated by the readers of the final version of the paper.
Pages 13-14: The geomorphological maps are great!
Sincerely
Reviewer 3 Report
COMMENTS AND SUGGESTIONS FOR AUTHORS
The present work is interesting, is well structured and assessed one proposal for the flood risk mitigation in the Upper Tanaro Valley in the Western Alps, in Italy. The paper is well explained and the results are well represented with plenty of tables and figures. More bibliographic citations could be added.
I give my suggestions for improving the manuscript, which should be done before publication of the paper.
- Introduction
- The introduction is well structured.
- Not too many bibliographical citations.
- The objective of the work is well defined.
- Setting
- Figure 1 is appropriate.
- In line 172, remove the description of the photo or add the picture.
- Methodologies
- Remove cite Nº25, in line 192.
- Figure 2 is appropriate but the visual quality of the maps should be improved.
- In equation 2, between lines 247-248 remove the comma.
- Results
- In Figure 4, the map legend shows classes that are not shown on the map in the geomorphology section. For example, superficial landslides, creep, rockfall & topping and DSGSD.
- The legend of the geomorphological maps (Figure 7) are generally ordered by the morphogenetic origin of the geomorphological forms and deposits (Structural, Fluvial, Karstic...) and their symbology must be in accordance with their origin. It´s suggested ordering the legend and changing the symbology/colors of the map.
- Figure 8 suggests the same comment as above. It should be noted that the symbology on these maps is much more appropriate. However, it still needs to be ordered.
- The rest of the figures and tables are appropriate.
- Discussion
- Are appropriate.
- Conclusions
- Are appropriate.

Round 2
Reviewer 1 Report
Dear Authors,
Thank you for your responses. However, many of them were not a real response, as the same mistakes remained, and still the locations of the the to-be-dams are not cleared. The quality of the paper did not become better. I am sorry for that.
